# A Method for Determining Ultimate Grouting Pressure for Reinforcement of Masonry Arch Dam with Mortar Deterioration: A Case Study

**DOI:** 10.3390/ma15103520

**Published:** 2022-05-13

**Authors:** Jia’ao Yu, Zhenzhong Shen, Liqun Xu, Chuankai He

**Affiliations:** 1State Key Laboratory of Hydrology-Water Resources and Hydraulic Engineering, Hohai University, Nanjing 210024, China; jiaaoyu@hhu.edu.cn (J.Y.); zhzhshen@hhu.edu.cn (Z.S.); 2The College of Water Conservancy and Hydropower Engineering, Hohai University, Nanjing 210024, China; 3Datang Hydropower Science & Technology Research Institute Co., Ltd., Nanning 530007, China; kaichuanhe@126.com

**Keywords:** masonry arch dam, mortar deterioration, reinforcement grouting, ultimate grouting pressure, grouting test

## Abstract

The deterioration of mortar has an adverse impact on the deformation and stress state of the masonry arch dam, after freeze-thaw cycles, in long-term operation. The purpose of this paper is to investigate the effect of reinforcement grouting on the stress of a thin masonry arch dam and propose a reasonable grouting method in the case of mortar deterioration. The determination of the ultimate grouting pressure is another main focus. The masonry material was generalized by combining a linear elastic model and the proportional weighted average under the condition of deterioration caused by freeze-thaw cycles. A series of analytical methods were proposed for the research of grouting effect on dam stress, based on which the ultimate grouting pressure is calculated in various cases. Results demonstrate that the dam tensile stress may exceed the allowable value in the following operation. Then, some recommended methods for the grouting layout and the estimation of grouting pressure were put forward by integrating the grouting field test with numerical analysis for reinforcement. The research conclusions might have a guiding significance for the reinforcement of similar projects.

## 1. Introduction

Since the 1950s, a large number of arch dams have been built in China due to the advantages of low cost and the convenient construction of masonry structure. By the end of 2000, there were about 1600 masonry arch dams built in China, accounting for about 90% of the total number of small and medium-sized arch dam projects [1]. These masonry arch dams have been in operation for decades, and different degrees of safety problems emerged recently due to the limitations of design capacity and construction funds. As a result, the normal operation of the project is easily affected, and there is even a threat to public safety [2].

Masonry arch dams are mainly composed of block stones and cement mortar. Cement mortar is in a thin-layer state, which mainly plays the role of bonding block stones and transferring stress. Due to the long-term influence of freeze-thaw cycles, water infiltration, high-speed scouring, and other factors, the mechanical properties of the internal mortar will gradually decrease, resulting in cracking and leakage. The deterioration rate of mortar is much higher than that of block stones, which may cause the damage of an arch dam in serious cases. In recent years, plenty of profound studies have been carried out on the deterioration of cement mortar [3,4,5,6]. For areas with cold winters and large temperature differences between day and night, mortar deterioration is dominated by the effect of freezing-thawing cycles [7,8,9]. However, the effect of mortar deterioration on a masonry arch dam, under the condition of freezing and thawing, is still remained uncertain.

For the old projects made by mortar masonry, if there are penetrating cracks owing to mortar deterioration, the cement grouting is generally utilized for reinforcement. The cement slurry is grouted into the dam body by relative equipment to fill the gaps inside the masonry dam. At the same time, the impervious body is formed through grouting, so the bearing capacity and integrity of the dam can also be further improved [10,11,12]. In the process of cement grouting, a reasonable layout scheme of grouting holes should be planned in advance, including the diameter of the grouting hole, distance between adjacent holes, drilling depth, and so on. Besides, the optimum grouting pressure is necessary to be determined. The higher the grouting pressure is, the more conducive it is to fill the cracks and gaps with slurry, so as to obtain better reinforcement effect. On the other hand, if the grouting pressure is too high, the artificial cracks are likely to be caused by the injection of slurry, especially for thin arch dams. There may be phenomena, such as oozing and fracturing, in this case. Thus, only by ascertaining the appropriate grouting pressure and maintaining stability in the construction process can the dam reinforcement effect be ensured.

There is no specific specification for the reinforcement grouting on a masonry arch dam presently. The existing standards of cement grouting mainly focus on foundation treatment, so the reinforcement grouting on the arch dam body is usually based on the method and experience of foundation grouting [13,14]. For example, in the topic of foundation anti-seepage treatment discussed in the fifteenth International Dam Conference in 1985, it was pointed out that the most viscous slurry and the maximum allowable pressure should be adopted in order to improve the quality of cement grouting, and the grouting pressure should be as stable as possible [13]. According to the grouting technical standards (EM1110-2-3506) proposed by the United States Army Corps of Engineers (USACE) [14], the grouting hole diameter is decided by the type and depth of the hole, with the minimum value of around 38–76 mm, and generally, high pressure is required in grouting, but the maximum value shall not exceed 3.0 MPa.

There are a series of conservancy industry specifications that introduced cement grouting methods in China as well [15,16]. For instance, the design specification for a stone masonry dam (SL25-2015) points out that, if the structure of masonry dam cannot meet the anti-seepage requirements, filling grouting shall be used for reinforcement to improve the compactness of materials and the anti-seepage performance of the dam body [15]. In addition, the technical specification for cement grouting of hydraulic structures (SL62-2014) offers corresponding design advice for construction technologies, such as bedrock consolidation grouting, joint grouting, and contact grouting, but no specific requirements are put forward for the construction technology of filling grouting [16].

In summary, the construction of reinforcement grouting for a mortar masonry dam is only based on engineering experience, and the construction methods have not been proven by numerical analysis. So far, there have been a lot of cases of reinforcing masonry arch dams by filling grouting in China [17,18]. According to the construction results of various projects, it is considered that the reinforcement grouting with cement slurry has a positive effect on eliminating the potential safety hazards of masonry dams and improving the reliability of project operation.

Therefore, a double curvature arch dam in eastern China is taken as the research case to study the reasonable method of reinforcement grouting for a mortar masonry arch dam, considering the effect of mortar deterioration, in this paper. Based on the finite element method (FEM), the influence of mortar deterioration, on the displacement and stress of the masonry arch dam, is numerically analyzed at first. Secondly, aiming at the problems of unclear specifications and immature technology in filling grouting for mortar masonry dams, a calculation method of the effect of reinforcement grouting on the stress of arch dam is put forward. Additionally, combined with the field test of grouting, the corresponding numerical analysis is carried out, and the selection of grouting pressure and the layout of grouting holes are also discussed. Last but not least, according to calculation results, suggestions are proposed for the reinforcement grouting of similar projects. The main conclusions of this paper may have an extensive reference value for the reinforcement of a large number of old masonry arch dams in China.

## 2. Description of the Studied Case

### 2.1. Project Overview

A hydropower station located in the south of Anhui Province, China, has comprehensive benefits such as power generation, flood control, and aquaculture, as shown in Figure 1. The reservoir of the hydropower station is a medium-sized reservoir with a multi-year regulation capacity, with the capacity of 3 × 10^7^ m^3^ at the normal storage level of 226.0 m. The water retaining structure is a mortar masonry double curvature arch dam with a maximum dam height of 55.0 m, which is a class 3 hydraulic structure. The downstream elevation and typical sections of the dam are shown in Figure 2, and the main design parameters of the arch dam are illustrated in Table 1.

Since the hydropower station completed in 1992, the arch dam has been in operation for about 30 years. At present, some safety problems have occurred in the dam. For example, mortar on the upstream dam surface have cracked and fallen off, and obvious leakage has occurred at the elevation of 190.00–208.00 m on the downstream surface. Therefore, it is necessary to consolidate the dam with reinforcement grouting.

### 2.2. Cement Grouting Test

#### 2.2.1. Test Position and Grouting Method

Before the formal reinforcement grouting, the cement grouting test was carried out on the downstream surface of the dam body, according to relative specification [16]. The test site was within the elevation range of 204.00–214.00 m near the left bank observation room on the downstream of the dam. The grouting holes were arranged in a quincunx shape on the downstream surface, with five rows in total, and the number of holes in each row, from top to bottom, is six, five, six, five and six, respectively. The maximum depth of grouting holes was 2.8 m, and the hole diameter was 40 mm. The grouting test site and test equipment are shown in Figure 3. The method of circulating grouting was adopted to keep the grouting pressure stable at 0.55 MPa.

#### 2.2.2. Water Pressure Test

A water pressure test shall be conducted before the grouting test [19]. After drilling and high-pressure cleaning, the grouting pipe was buried for the water pressure test. There was no obvious water seepage after drilling. The water permeability of the grouting hole is calculated according to Equation (1)
(1)$$q=\frac{Q}{PL}$$
where *Q* is the total water injection in test section per unit time; *P* is the pressure acting on the test section; *L* is the length of test holes. The water pressure test results show that the water permeability of the test area is between 0.29–0.37 Lu.

#### 2.2.3. Grouting Effect

Portland cement was selected as the grouting material, and the grouting cement ratio was 2:1 at first, and then, the ratio was gradually increased to 1:1 subsequently. The dam downstream surface, before and after the grouting test, is shown in Figure 4. In Figure 4a, we can see that the phenomena of moisture and precipitation of white solids around the mortar layer between block stones are pretty obvious, which indicates that the mortar has deteriorated in a serious degree. Considering that the mortar layers are connected in the horizontal direction and not in the vertical direction, the cracks and cavities are mainly horizontal. Therefore, the grouting holes are arranged at the severely deteriorated part of the left downstream surface of the dam, and each row is arranged horizontally in the horizontal mortar layer, as shown in Figure 4b. It can be seen that the grouting of cement significantly filled the cavities and cracks in the dam body, and the compactness of dam body and the degree of cementation between blocks were effectively increased.

The grouting holes for arch dam reinforcement can only be drilled straightly and cannot be drilled along the curved shape of the arch dam. Additionally, because the thickness height ratio of the arch dam is small and the dam crest is narrow, it is impossible to drill and grout vertically from the dam crest down, so the method of horizontal grouting from the downstream surface to upstream was adopted. Since the grouting with the packing gland method is adopted, the pressure in all directions in the closed hole is the same during the grouting process. Therefore, the influence law of horizontal and vertical grouting methods on the arch dam can be used for reference.

## 3. Simulation Method for Stress and Deformation of Arch Dam

### 3.1. Constitutive Model of Materials

On the basis of the main mechanical parameters of various masonry, proposed by the design specifications for a stone masonry dam (SL25-2015), the materials of the dam body and foundation are all regarded as linear elasticity [15]. Combined with similar engineering experience, the specific calculation parameters of the dam body and foundation are determined, respectively, as shown in Table 2.

### 3.2. Mortar Deterioration under Freeze-Thaw Cycle Condition

According to the on-site safety inspection, there was leakage in the downstream of the dam body and abutment, and a large amount of calcium was precipitated in the mortar. Since the mortar is cement-based, the precipitated calcium-containing compounds are mainly calcium hydroxide, which forms calcium carbonate after contacting with carbon dioxide in the air. Therefore, the mechanical properties of the dam materials are reduced, and the impact of mortar deterioration on dam safety needs to be considered. The arch dam is located in the temperate zone, so there is no long-term freezing or serious weathering. Due to the small height and low water level of the arch dam, the scouring and infiltration of water flow are not obvious. The reservoir water quality is good, which has little effect on mortar deterioration. On the other hand, the temperature difference between day and night in winter at the dam site is large, so it is considered that freeze-thaw is the main factor causing the deterioration of cement mortar [20].

The relationship between the elastic modulus of mortar and the number of freeze-thaw cycles is as follows in Equation (2)
(2)$$E_{n}=a\times E_{0}\times \exp(-b\times n)$$
where $E_{0}$ is the elastic modulus of mortar without freeze-thaw; *n* is the number of freeze-thaw cycles; $E_{n}$ is the elastic modulus of mortar after *n* times of the freeze-thaw cycle during tests; *a* and *b* are the test parameters. Based on the regression analysis proposed by Yao et al. [20], a≈1.01172, b≈0.00314.

However, in the actual engineering operation environment, it is difficult to determine the exact number of times that buildings are subjected to freezing and thawing. Therefore, for the analysis of frost resistance durability of masonry structure, the equivalent number of freeze-thaw cycles proposed by Yin et al. is utilized to represent the number of freeze-thaw cycles *n* in this paper, which is as follows [21,22,23]:(3)$$\left[N\right]=N_{-15–+5{}^{{}^{\circ}}C}+0.1N_{-10–+5{}^{{}^{\circ}}C}+0.01N_{-5–+5{}^{{}^{\circ}}C}$$
where [*N*] is the equivalent number of freeze-thaw cycles; *N*_−15–+5°C_ is the number of freeze-thaw cycles between −15 °C and +5 °C; *N*_−10–+5°C_ is the number of freeze-thaw cycles between −10 °C and +5 °C; *N*_−5–+5°C_ is the number of freeze-thaw cycles between −5 °C and +5 °C.

Based on the statistics of meteorological data of the dam site over the years, the number of freeze-thaw cycles of the dam site area in each month are obtained and shown in Table 3.

Combined with Equations (2) and (3), the equivalent number of freeze-thaw cycles and corresponding deterioration degrees of the arch dam, under different operation years, are calculated and shown in Table 4. It can be seen from the table that the elastic modulus of mortar is basically linear with the number of freeze-thaw cycles, so the current elastic modulus of mortar of the arch dam can be estimated by the interpolation method.

### 3.3. Load Calculation and Combination Method

#### 3.3.1. Load Combination

The bottom thickness and the height of the arch dam are, respectively, 13.331 m and 55.0 m, and the corresponding thickness height ratio is 0.242, so this arch dam is a medium thick one. In structure analysis, the influence of upstream and downstream hydrostatic pressure and uplift pressure shall be taken into account at first. The vegetation in the upstream of the dam site area is good, and as a result, the sediment pressure can be ignored. According to relevant regulations in the design specifications for stone masonry dams (SL25-2015) [15], the different combinations of hydrostatic pressure, temperature load, dam self-weight, and uplift pressure are considered.

#### 3.3.2. Calculation Method of Temperature Load

The temperature load, exerted on the arch dam during operation, is mainly caused by the change of ambient temperature, which is calculated according to the formulas in the specification. The multi-year average temperature in the dam site area, of 15.4 °C, is taken as the arch sealing temperature, and the difference between the arch sealing temperature and the multi-year monthly average minimum and maximum temperature shall be adopted for the normal temperature drop and temperature rise, respectively. The multi-year monthly average minimum and maximum temperatures are 3.0 °C and 27.3 °C, respectively.

The temperature of the underwater part of the dam upstream surface is calculated as follows [24]:(4)$$T_{w}\left(y,\tau \right)=T_{wm}\left(y\right)+A_{w}\left(y\right)\cos\left\{\frac{2\pi }{p}\left[\tau -\tau_{0}-\varepsilon \left(y\right)\right]\right\}$$
where $T_{w}\left(y,\tau \right)$ is the multi-year average water temperature at water depth *y* in month *τ*; *τ* is the time variable; *τ*_0_ is the initial phase, usually taken as 6.5; *p* is the temperature change period, generally taken as 12; *ε*(*y*) is the phase difference, as shown in Equation (5); $A_{w}\left(y\right)$ is the multi-year annual variation of temperature at water depth *y*, which is calculated by Equation (6); $T_{wm}\left(y\right)$ is the multi-year annual temperature at water depth *y*, as shown in Equation (7).
(5)$$\varepsilon \left(y\right)=2.15-1.30e^{-0.085y}$$
(6)$$T_{wm}\left(y\right)=\left(7.77+0.75T_{am}\right)e^{-0.01y}$$
(7)$$A_{w}\left(y\right)=\left(2.94+0.778A_{a}\right)e^{-0.025y}$$
where *T**_am_* and *A**_a_* are the multi-year monthly average temperature and the annual average temperature variation, as shown in Equations (8) and (9), respectively,
(8)$$T_{am}=\frac{1}{12}\sum_{i=1}^{12}T_{ai}$$
(9)$$A_{a}=\left(T_{a7}-T_{a1}\right)/2$$
where *T**_ai_* is the annual average temperature in the month *i*.

In addition, the annual variation of the downstream surface temperature of the arch dam can be calculated according to Equation (10).
(10)$$T_{a}=T_{am}+A_{a}\cos\left[\frac{2\pi }{p}\left(\tau -\tau_{0}\right)\right]$$

At normal storage level, based on the multi-year monthly average minimum and maximum temperatures, the relation curves between water temperature and water depth in January and July are obtained by Equations (4)–(10), as shown in Figure 5.

### 3.4. Control Standard of Dam Stress

Based on the design specifications for a stone masonry dam (SL25-2015) [15] and the design specifications for concrete arch dams (SL282-2018) [24], when the stress of arch dam is calculated by the finite element method, the allowable stress control indexes of the arch dam are shown in Table 5.

## 4. Simulation Method of Grouting Effect

### 4.1. Effect of Grouting on Dam Surface

During the reinforcement grouting construction, if the grouting scheme is not arranged correctly or the grouting pressure cannot be controlled, it is probable to cause damage to the dam body. In order to ensure that the artificial penetrating cracks leading to the dam upstream surface will not be produced during grouting, it is necessary to study the relationship between the grouting borehole position, grouting pressure, and the maximum tensile stress of the dam body. In this paper, the three-dimensional finite element method is used for simulation analysis. Firstly, the location of the grouting hole section is preliminarily determined, and the geometric size of the grouting hole is ignored. Then, the method of “replacing hole with line” is adopted to distribute grouting holes to element nodes. Therefore, the grouting pressure can be transformed into concentrated forces and applied to the element nodes at the corresponding positions through integration. Because the upstream surface of the arch dam is close to the water, the effect of grouting on its stress needs to be focused. Thus, the direction of these concentrated forces is set to point upstream to simulate the most adverse effect of grouting on the upstream stress. When the position of the grouting hole changes, the node under the action of concentrated force, transformed by grouting pressure, also changes. The schematic diagram of the grouting impact simulation is shown in Figure 6.

Thus, the equivalent concentrated forces, acting on the element nodes at the termination and middle of the grouting hole section, are calculated by Equations (11) and (12), respectively,
(11)$$F_{1}=\pi rpl$$
(12)$$F_{2}=2\pi rpl$$
where *r* is the radius of grouting hole; *l* is the height of element; *p* is the grouting pressure.

### 4.2. Determination of Ultimate Grouting Pressure Based on Pure Arch Method

In addition to the dam stress, it is also critical to study the influence of grouting on the local stress around the grouting hole to further control the ultimate grouting pressure. Based on the theory of the pure arch method, the typical arch ring section is selected at the grouting position to establish a two-dimensional finite element model in this paper. The corresponding hydrostatic pressure, grouting pressure, and temperature load are applied on the arch ring model, and the left and right sides of the arch ring model on dam abutment are set as fixed constraints. By changing the parameters such as grouting condition, grouting hole position, grouting hole diameter, and elastic modulus of dam material, the ultimate grouting pressure under different conditions can be calculated, and the law of construction can be summarized. The calculation diagram of the pure arch method is shown in Figure 7.

## 5. Simulation Results

### 5.1. The Finite Element Model

There have been a series of research results on masonry modeling [25,26]. In a mortar masonry arch dam, the number of block stones is huge, and the size is random, so there is no accurate modeling method at present. Therefore, the dam body is regarded as homogeneous material for modeling and analysis. The three-dimensional (3D) finite element model of the arch dam was established according to the actual shape, with 126,136 elements and 130,825 nodes in it, as shown in Figure 8a. The X-axis is along the river direction, the Y-axis is perpendicular to the river direction, and the Z-axis is vertically upward, with the real elevation as its coordinates. The calculation scope included arch dam, dam abutment mountain, and dam foundation rock mass, and the auxiliary structures such as built-in corridor, water conveyance tunnel, and flip bucket were ignored.

Besides, the two-dimensional (2D) local model of an arch ring at the reinforcement grouting position was also established to determine the ultimate grouting pressure, under different conditions, by pure arch method, and the local finite element model is shown in Figure 9. The meshes in the area around the grouting hole were locally densified by transition technology.

### 5.2. Dam Displacement and Stress Considering the Freeze-Thaw Degradation of Mortar

Generally, the deterioration degree of stone in masonry is small and can be ignored. Therefore, it is assumed that the elastic modulus of stone remains unchanged with the increase in project operation time, while the elastic modulus of mortar gradually decreases under the influence of deterioration, and the deterioration is only related to the equivalent freeze-thaw cycle times of mortar.

Based on empirical data, 0.32 cubic meters of mortar is required for each cubic meter of masonry. Thus, the mortar content and the comprehensive elastic modulus of the dam body can be estimated according to the proportion. Until now, the arch dam has been in service for 29 years since the completion of the construction. Referring to Table 4, the equivalent freeze-thaw cycle times and corresponding deterioration degree were calculated by linear interpolation and determined to be 116 and 29.1%, respectively. In addition, the sensitivity analysis of dam deformation and stress under different deterioration degrees was carried out. Combined with the material parameters proposed by the design specification for stone masonry dams (SL25-2015), the calculation schemes were shown in Table 6.

Ignoring the effect of mortar deterioration and reinforcement grouting, the above calculation schemes were adopted to calculate the stress and deformation of the arch dam under the normal storage level, and the contours of dam displacement were obtained and illustrated in Figure 10. It is noteworthy that the displacement caused by gravity was cleared before exerting other loads to avoid the effect of sectional placing, during the construction phase, on dam displacement.

The calculation results shown in Figure 10 indicate that the distributions of dam displacement were symmetrical around the arch crown section, and the displacement along the river primarily points to downstream, which conforms to the general law. Besides, the stress contours are shown in Figure 11.

On the other hand, considering the different degrees of mortar deterioration, the stress and displacement calculations of the arch dam were also carried out. The comparisons of maximum values of displacement and stress were shown in Table 7. Based on the calculation results, the value of displacement components and the tensile stress of dam body increased with the increase in mortar deterioration degree. By contrast, the absolute value of the dam compressive stress declined as time went on. When the operation time exceeds 40 years, the deterioration degree of mortar will reach 39.1%, and the maximum tensile stress of the dam body (1.51 MPa) will exceed the allowable tensile stress (1.50 MPa). The displacement along the river and the first principal stress of the arch dam are predicted to increase by 14.9% and 7.1%, respectively, which may have an adverse impact on the structure safety. Therefore, engineering measures need to be taken for reinforcement of the dam.

### 5.3. Effect of Grouting on Stress of Dam Upstream Surface

Ignoring the geometric dimension of grouting holes, the method of “replacing hole with line” was adopted to convert the grouting pressure into concentrated force, and the concentrated force was applied to the corresponding nodes to study the effect of grouting on the stress of the dam upstream surface. It is assumed that the perpendicular grouting line is arranged on the arch crown section. Firstly, the concentrated force was applied to the element nodes of the dam downstream surface. The stress distributions of the dam body near the grouting area, under the conditions of ignoring and considering grouting pressure, are shown in Figure 12. Since the stress distribution of the dam body is symmetrical to the arch crown section, only half of the grouting area was displayed.

Respectively, on the other hand, 2 times, 2.5 times and 3 times of the upstream water head were taken as the grouting pressure, and the position of grouting holes was changed for sensitivity analysis. The maximum values of the first principal stress on the dam upstream surface, under different conditions, were summarized in Table 8.

It can be concluded that the reinforcement grouting inside the dam body will have a certain impact on the stress of the dam surface, but the impact is very small, which can be ignored. Besides, the stress maximum value of the dam upstream surface is far less than the allowable tensile stress, so it is not likely for cracks to develop from the upstream surface to the downstream. Therefore, in the grouting process, the ultimate grouting pressure should be controlled by the condition that the local maximum stress around the grouting hole section ought to be less than the allowable tensile stress. Then, the determination of ultimate grouting pressure and the influence of grouting on the local stress around grouting holes were discussed below.

### 5.4. Determination of Ultimate Grouting Pressure under Different Conditions

#### 5.4.1. Local Effect of Stress around Grouting Hole

Firstly, it is assumed that the grouting hole is arranged in the middle of the symmetry axis of the arch ring. Under the condition of a normal storage level, the corresponding water pressure was applied on the upstream surface, and 2 times of the upstream water head was taken as the grouting pressure for finite element calculation. The stress contours of the whole arch ring and the local area around the grouting hole are shown in Figure 13 and Figure 14, respectively.

As we can see from Figure 13 and Figure 14, the grouting pressure exerted will only change the stress distribution in the range of 5 times that of the hole diameter near the grouting hole, and the effect on other positions can be ignored, which conforms to the calculation results illustrated in Figure 12 and Table 8. To determine the ultimate grouting pressure, it is necessary to analyze the principle between the local maximum tensile stress and the grouting pressure at different locations.

#### 5.4.2. Determination of Ultimate Grouting Pressure

The grouting hole, in the middle of the symmetry axis of the arch ring, was taken as an example for the determination of ultimate grouting pressure at this position. By changing the grouting pressure, the extreme values of local stress around the grouting hole can be calculated, as shown in Table 9. According to the above results and the method of fitting, the relationship curves between the extreme values of local stress around the grouting hole and grouting pressure were obtained, as illustrated in Figure 15. It can be seen that the local stress extreme value around the grouting hole basically conforms to the linear relationship with the grouting pressure.

It is assumed that, if the local extreme value of the first principal stress reaches the allowable tensile stress, the tensile failure will occur around the grouting hole; if the local extreme value of the third principal stress reaches the allowable compressive stress, the compressive damage will occur. Combined with the relationship curves in Figure 15, the allowable tensile stress (1.50 MPa) was utilized as the control condition, and the ultimate grouting pressure (0.78 MPa) was determined.

Similarly, the upstream water level of the arch ring was adjusted to the normal storage level and the design flood level, respectively. Besides, the location of the grouting hole was also changed, which is expressed by the ratio of d to T. Then, the ultimate grouting pressures in different grouting locations and at different upstream water levels were summarized. By quadratic curve fitting, the relationship curves between the ultimate grouting pressure and the grouting location, under different water level conditions, were calculated and expressed as Equations (13)–(15).
(13)$$P=\left(2.45\pm 0.29\right)r^{2}+\left(-1.69\pm 0.30\right)r+\left(1.23\pm 0.07\right),R^{2}=0.971$$
(14)$$P=\left(3.09\pm 0.34\right)r^{2}+\left(-1.90\pm 0.35\right)r+\left(0.92\pm 0.08\right),R^{2}=0.981$$
(15)$$P=\left(3.47\pm 0.40\right)r^{2}+\left(-1.97\pm 0.41\right)r+\left(0.65\pm 0.09\right),R^{2}=0.983$$
where *r* is the value of d/T; *P* is the corresponding ultimate grouting pressure.

According to Figure 16, under different water storage level conditions, the relationship between the grouting position and the corresponding ultimate grouting pressure accord well with a quadratic parabola, and the highest pressure that can be adopted for grouting appears on the downstream side of the arch ring, while the lowest pressure that can be adopted appears when d/T is equal to 0.3. In addition, the higher the upstream water level, the lower the grouting pressure can be adopted.

### 5.5. Simulation Results of Grouting Test

Through Boolean calculation, the corresponding position of the finite element model shown in Figure 8a was modified according to the real size of the grouting hole in the grouting test, as shown in Figure 8b,c. The method of circulating grouting was adopted, so it can be assumed that the pressure in all directions in the grouting hole was kept stably at 0.55 MPa. The simulation result of local stress in the grouting test area is shown in Figure 17.

In the grouting test area, the horizontal sections were cut along the center line of each row of grouting holes to study the stress distribution in these sections. Stress contours of each section are summarized in Figure 18.

Since the extreme values of local stress near the grouting holes are still less than the allowable stress, the grouting pressure can be increased within a certain range. The ultimate grouting pressure is still controlled by the local tensile stress. When the grouting pressure is equal to 2.0 MPa, the tensile stress distributions of each section were simulated and shown in Figure 19.

Similarly, the local stress around each grouting hole, when the grouting pressure is equal to 5.0 MPa, can also be estimated. It can be seen from Figure 18 and Figure 19 that, under the same grouting pressure, the lower the altitude of the grouting holes and the further away from the crown cantilever, the smaller the local maximum tensile stress around it. Additionally, with the increase in grouting pressure, the position of local maximum tensile stress will be transferred from the periphery of the orifice to the bottom of the hole. The three-dimensional fitting surface of maximum tensile stress in the test area, under different grouting pressure, was illustrated in Figure 20. Based on the simulation results, it can be seen that the local stress extreme value around each hole still approximately conforms to the linear relationship with the grouting pressure in horizontal grouting from downstream to upstream.

Referring to the method for determining the ultimate grouting pressure proposed previously, the 3D fitting surface of ultimate grouting pressure in the test area was obtained, as shown in Figure 21.

Based on Figure 21, the lower the altitude of the horizontal grouting holes, and the closer they are to the dam abutment on the left bank, the greater the grouting pressure that can be utilized. At the location that is 11.0 m away from the crown cantilever and 205.0 m at the altitude, the ultimate grouting pressure is able to be calculated, which is about 5.5 MPa.

### 5.6. Sensitivity Analysis

#### 5.6.1. Sensitivity Analysis of Grouting Hole Diameter

Under the condition of a normal storage level, the diameter of grouting holes was changed, and the corresponding ultimate grouting pressure at different positions was calculated, as shown in Table 10. Then, the effect of the grouting hole diameter on ultimate grouting pressure was illustrated in Figure 22, based on Table 10. Thus, it can be learned that the larger the diameter of the grouting hole on the upstream and downstream surface of the arch dam, the higher the ultimate grouting pressure can be adopted, while the effect of the diameter on the grouting pressure is slight in the middle of the arch ring. 

#### 5.6.2. Sensitivity Analysis of Ambient Temperature Reduce

Since the grouting test was carried out in early spring, the effect of ambient temperature change on the grouting pressure is necessary to be considered. The temperature change in winter was regarded as the normal temperature drop load, and the temperature of the upstream and downstream surfaces of the arch ring was taken, respectively, as the conforming water temperature and ambient temperature (3.0 °C). The water temperature at the depth of the arch ring was calculated to be 8.5 °C by Equation (4). The initial temperature inside the arch ring is the arch sealing temperature of 15.4 °C. In summary, the temperature boundary conditions in the analysis of the arch ring temperature field are shown in Figure 23.

After calculation, considering temperature drop, the simulation results of the ultimate grouting pressure at various positions are shown in Table 11.

The effect of reduced ambient temperature on ultimate grouting pressure was demonstrated in Figure 24, based on Table 11. According to Figure 24, at the location where d/T equals to 0.3, the grouting pressure, in the case of reduced ambient temperature, is roughly the same as that ignoring temperature load. On the upstream side of this location, the ultimate grouting pressure will increase after considering the temperature reduction. On the contrary, the ultimate grouting pressure that can be adopted on the downstream side of the arch dam decreases by considering the temperature reduction.

#### 5.6.3. Sensitivity Analysis of Elastic Modulus of Dam Material

The elastic modulus of arch ring material was changed to analyze the influence on the ultimate grouting pressure. Through simulation, it is found that the change of modulus has little effect on the ultimate grouting pressure, which is reasonable to not be taken into account. The calculation results are omitted in this paper. Therefore, in the process of determining the ultimate grouting pressure, the influence of a material’s elastic modulus change, caused by different deterioration degrees, is ignored.

## 6. Discussion

### 6.1. Effect of Reinforcement Grouting on Masonry Arch Dam

Compared with the effect of grouting on the upstream surface of the dam, the influence of grouting on the local stress distribution around the hole is far more obvious. This phenomenon can be explained by the Saint–Venant principle, which indicates that, if a force is applied on a small part of the boundary of an object, the stress distribution near the boundary will change significantly, but the effect on the distance is negligible. Thus, this paper mainly concentrates on the analysis of the grouting effect on local stress. 

Based on relative calculations, in the case that the grouting pressure is between two and three times that of the upstream water pressure, the maximum value of local tensile stress around the hole is linearly and positively correlated with the grouting pressure. This law is practical in both vertical and horizontal grouting. Therefore, the method of inferring the ultimate grouting pressure of a certain grouting hole at its corresponding position through the relationship curve between grouting pressure and local maximum tensile stress can be considered as reasonable. On the other hand, the influence of reinforcement grouting on the increase in tensile stress, in the middle of arch ring, is greater than that in the upstream side. The cause of this situation may be that the compressive stress on the upstream surface is greater than that in the middle of the arch ring. The compressive stress near the upstream surface can offset the tensile stress caused by grouting. Therefore, the applicable grouting pressure near the upstream surface is greater than that in the middle. 

### 6.2. Comparison of Simulation and Test Results with Current Grouting Specifications

Some provisions in current specifications can indeed be applied to the reinforcement grouting for the masonry arch dam. For example, according to the field grouting test, the higher the grouting pressure and the thicker the slurry used in the construction, the easier it is to inject the slurry into the arch dam, which is in accord with the topic of anti-seepage treatment discussed in the 15th International Dam Conference. However, the simulation results also show that some current grouting regulations are not perfect. For instance, it can be seen from Figure 16 that the vertical grouting pressure at some positions can only be 0.3 MPa under design flood level, while the grouting pressure can even exceed 5.0 MPa when horizontal grouting is adopted at some positions, according to Figure 21. The grouting technical standards (EM1110-2-3506) only stipulates that the maximum value of grouting pressure shall not exceed 3.0 MPa without taking specific situations into account, which is pretty ambiguous. 

As a result, pointing at the shortcomings of existing grouting specifications, this paper puts forward corresponding recommendations of reinforcement grouting for masonry arch dams.

### 6.3. Recommends of Reinforcement Grouting for Masonry Arch Dam

First of all, numerical simulations shall be carried out in combination with the actual situation to calculate the ultimate grouting pressure, and the allowable range of grouting pressure can be preliminarily judged. Then, the grouting field tests are supposed to be conducted. The grouting pressure can be gradually increased within the allowable range if the slurry is difficult to be injected into the dam body. If new splitting cracks appear during the test, the grouting shall be stopped immediately, and the test shall be carried out again after reducing the grouting pressure. The slurry for grouting should be as thick as possible. Otherwise, the thinner the slurry, the less solid matter formed after solidification. Therefore, the effect of filling and consolidation will be worse if the slurry is too diluted.

Secondly, the selection of the grouting hole diameter and the arrangement of the grouting scheme shall be fully considered before the grouting test. The grouting hole diameter is reasonable in the range of 40 mm to 80 mm. When vertical grouting is conducted near the upstream and downstream surfaces of the arch dam, the hole diameter can be properly increased to increase the grouting pressure. The grouting holes should be arranged in a quincunx shape, and the distance between adjacent grouting holes should not be less than five times of the hole diameter.

Thirdly, the grouting pressure is mainly determined by the arch dam shape, grouting position, and upstream water pressure. The higher the upstream water level during grouting, the smaller the ultimate grouting pressure. Under the same condition of upstream water level, when horizontal grouting is utilized, the lower the elevation, the greater the ultimate grouting pressure. During vertical grouting, the pressure of grouting on both upstream and downstream sides of the arch dam can be greater than that in the middle position. Therefore, it is recommended to carry out grouting on the downstream and upstream sides of the dam at first, and the reinforcement grouting in the middle part should be conducted after the slurry solidifies to ensure construction safety. Additionally, at the locations close to the arch dam abutment, the pressure that can be adopted for grouting is greater than that near the arch crown. There are great differences under different conditions, so the numerical analysis of grouting pressure shall be conducted for specific situations.

Last but not least, the impact of corresponding temperature changes in the construction season on reinforcement grouting is necessary to be considered. If grouting is conducted in winter, the compressive stress on the upstream surface is the largest, and the downstream surface will be in tension; if grouting is conducted in summer, the downstream surface compressive stress is the largest, and the upstream surface will be in tension. Due to the ultimate grouting pressure being controlled by tensile stress, the grouting construction near the downstream surface in winter and near the upstream surface in summer is obviously of the most adverse impact. Therefore, in this case, the grouting pressure shall be appropriately reduced to ensure safety when grouting is carried out at the downstream side of the arch dam.

### 6.4. Implications

The influence of mortar deterioration on the masonry arch dam was analyzed by a simplified method, and then, the related problems of reinforcement grouting, in order to make up for the adverse impact of mortar deterioration, are studied in this paper. The analysis method for the determination of ultimate grouting pressure has great guiding significance for the reinforcement construction of water-related projects, such as arch dams, gravity dams, and even channels, with masonry structures that have been in operation for many years and for the projects with mortar deterioration problem caused by the freeze-thaw cycle, especially.

However, the masonry material was generalized by combining the linear elastic model and the simple proportional weighted average method in this paper, which cannot accurately reflect the real characteristics of the structure. Further work will be necessary to better understand the mechanical behavior of this special material. For example, the block stones can be considered as quasi-brittle materials, and the reasonable interface model can be used to simulate the cohesive function of mortar in masonry structures. 

## 7. Conclusions

Based on numerical analysis and field tests, the relative questions of reinforcement grouting on a masonry arch dam under the condition of mortar deterioration is discussed, and the main conclusions are as follows:(1)The raise of mortar deterioration degree will lead to the value increase in the displacement components and the tensile stress of dam body, as well as the absolute value decrease in the dam compressive stress. The influence of mortar deterioration on the displacement along the river and the tensile stress is more obvious. It is predicted that the maximum tensile stress of the dam body may exceed the allowable tensile stress for another ten years of operation. As a consequence, some engineering measures are necessary to be taken for the reinforcement.(2)The analytical methods are proposed for the research of the grouting effect on dam stress. In detail, the influences of grouting on the upstream dam surface and the local area around the grouting hole, respectively, were investigated by the “replacing hole with line” method and the pure arch method. Additionally, the effect of horizontal grouting on the construction area was discussed by utilizing a three-dimensional finite element model, according to the actual situation of grouting test. The ultimate grouting pressure, in different positions and situations, can be determined as the basis of the effect of grouting on dam stress.(3)Combined with the grouting field test and simulation results, the research was conducted by comparing to present grouting specifications. The recommended methods for the grouting layout and the determination of grouting pressure were put forward. The simulation methods proposed and the research conclusions in this paper may have good application significance in the reinforcement construction for similar projects.

## Figures and Tables

**Figure 1 materials-15-03520-f001:**
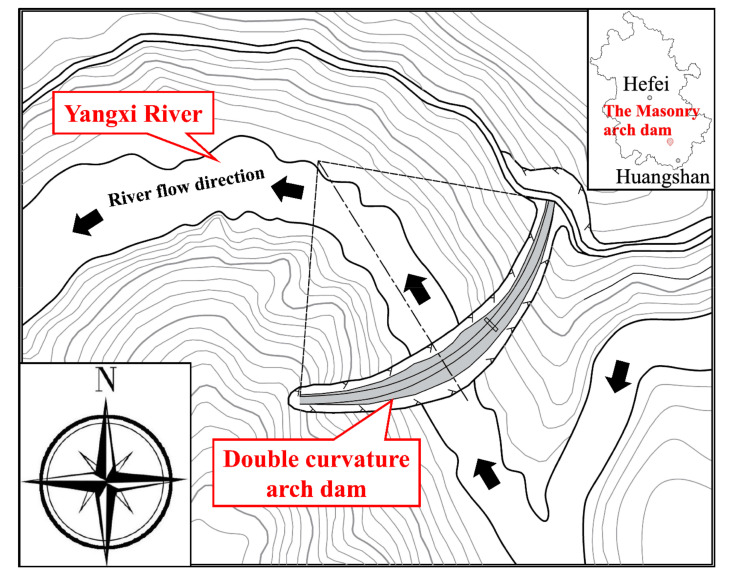
Location of the masonry arch dam.

**Figure 2 materials-15-03520-f002:**
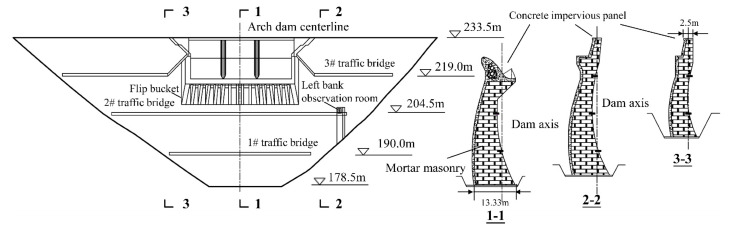
Downstream elevation and typical sections of the arch dam.

**Figure 3 materials-15-03520-f003:**
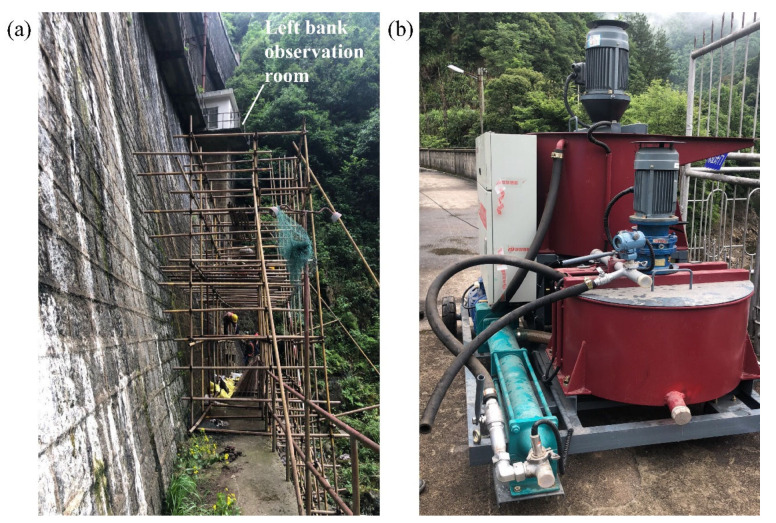
Grouting site and test equipment: (**a**) project site; (**b**) grouting test equipment.

**Figure 4 materials-15-03520-f004:**
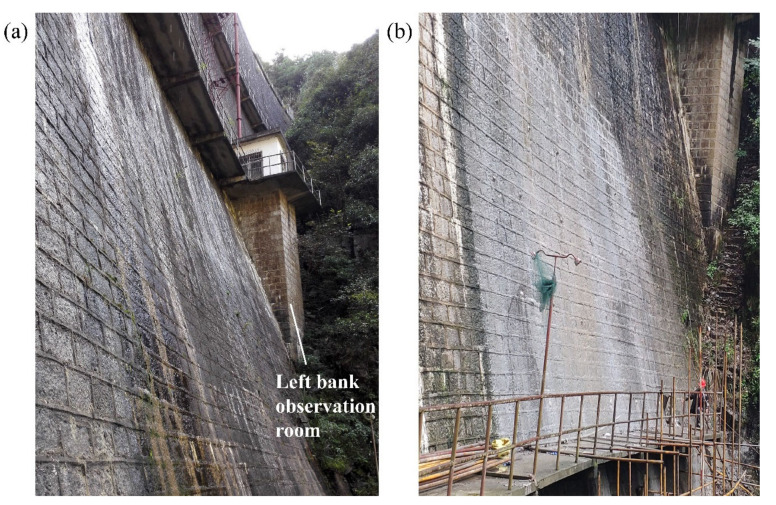
Comparison of dam downstream surface before and after the grouting test: (**a**) before grouting test; (**b**) after grouting test.

**Figure 5 materials-15-03520-f005:**
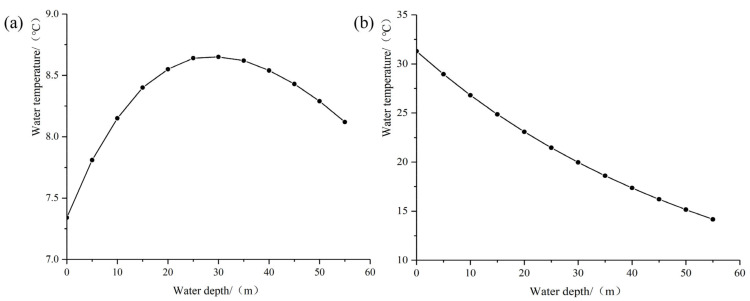
Relation curves between water temperature and water depth in different months: (**a**) July; (**b**) January.

**Figure 6 materials-15-03520-f006:**
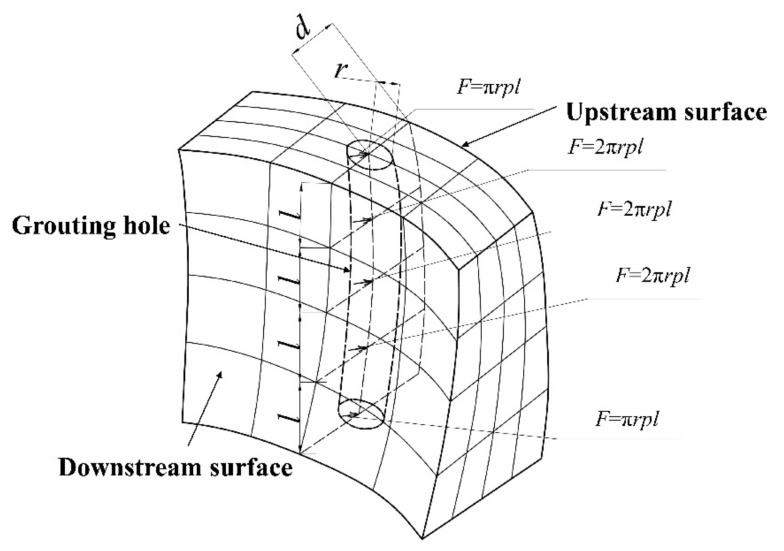
Schematic diagram of simulation of the effect of grouting on dam upstream surface.

**Figure 7 materials-15-03520-f007:**
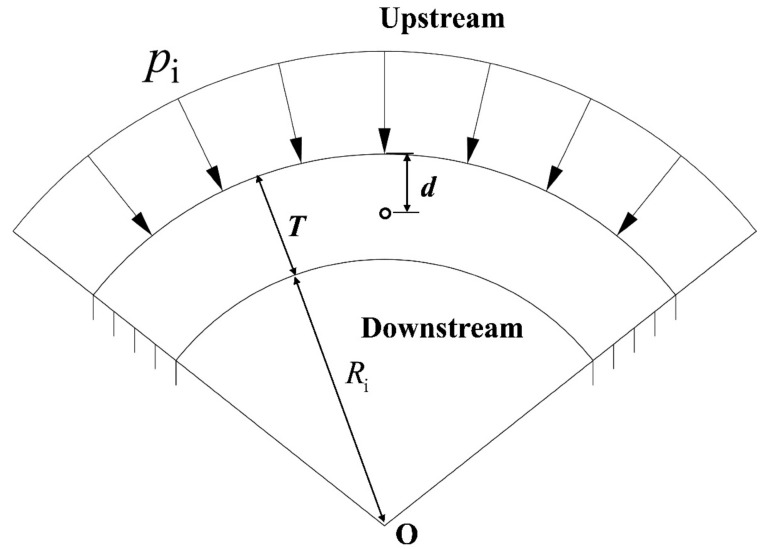
Calculation diagram of ultimate grouting pressure based on pure arch method.

**Figure 8 materials-15-03520-f008:**
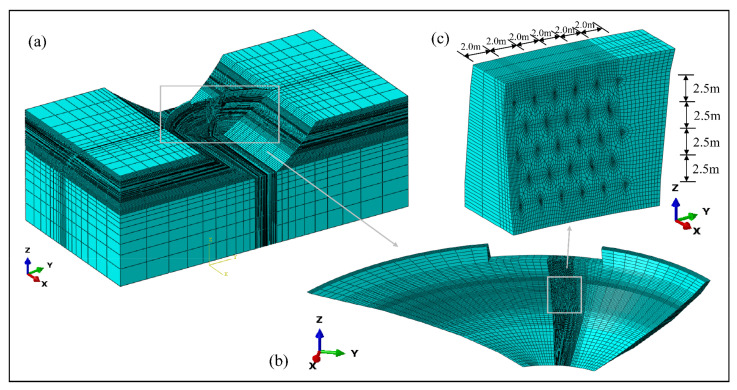
Three-dimensional finite element model of arch dam: (**a**) Overall model; (**b**) dam body; (**c**) Grouting test area.

**Figure 9 materials-15-03520-f009:**
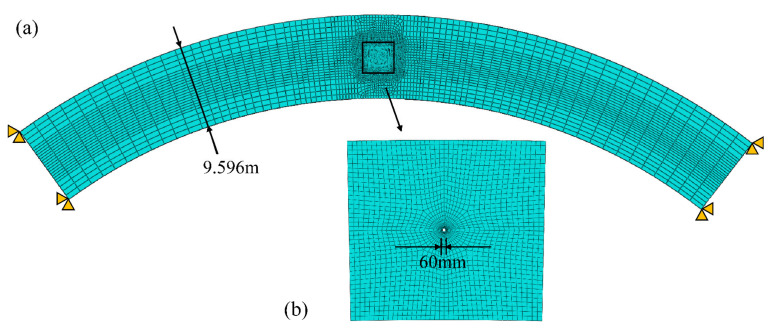
(**a**,**b**) Two-dimensional local finite element model of the arch ring.

**Figure 10 materials-15-03520-f010:**
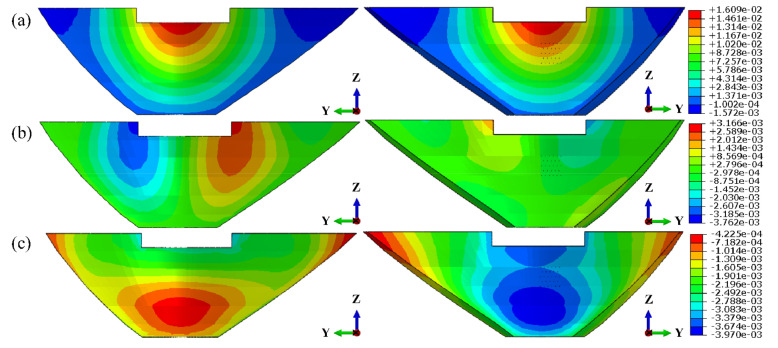
Contours of dam displacement (ignore grouting pressure and mortar deterioration): (**a**) displacement along the river; (**b**) displacement perpendicular to the river; (**c**) settlement. (unit: m).

**Figure 11 materials-15-03520-f011:**
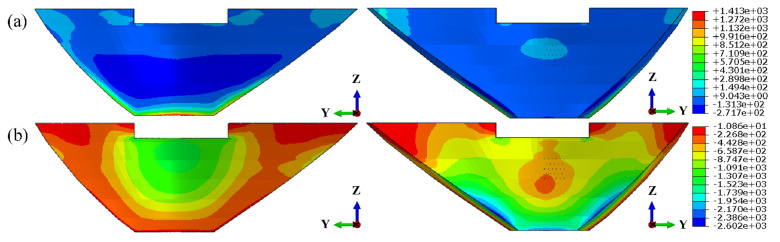
Contours of dam stress (ignore grouting pressure and mortar deterioration): (**a**) 1st principal stress; (**b**) 3rd principal stress. (unit: kPa).

**Figure 12 materials-15-03520-f012:**
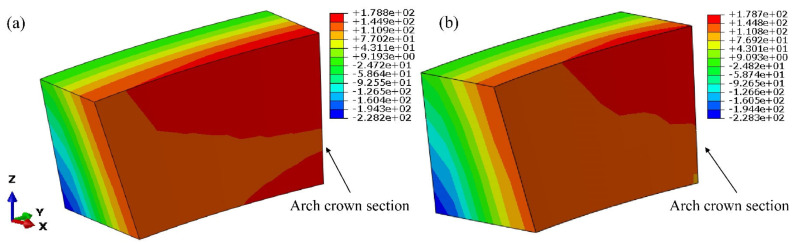
Local stress comparison of the grouting effect on the upstream surface: (**a**) ignore grouting pressure; (**b**) consider grouting pressure. (unit: kPa).

**Figure 13 materials-15-03520-f013:**
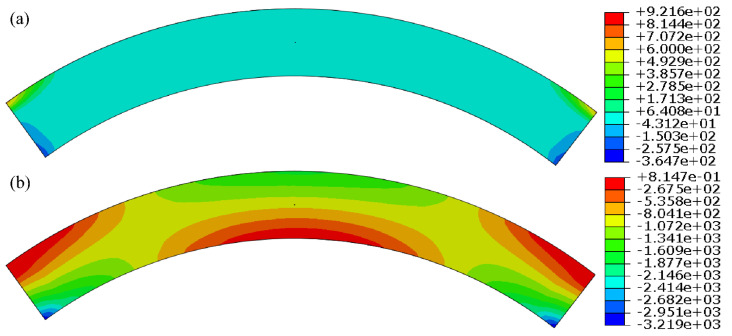
Stress contours of arch ring under grouting: (**a**) 1st principal stress; (**b**) 3rd principal stress. (unit: kPa).

**Figure 14 materials-15-03520-f014:**
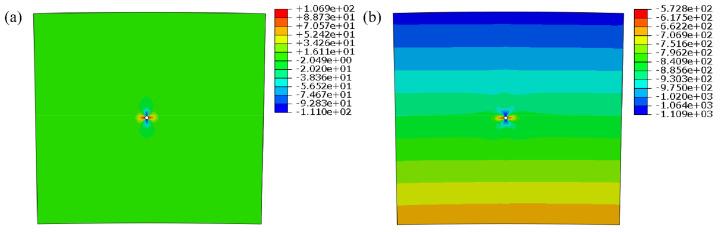
Local stress contours around the grouting hole: (**a**) 1st principal stress; (**b**) 3rd principal stress. (unit: kPa).

**Figure 15 materials-15-03520-f015:**
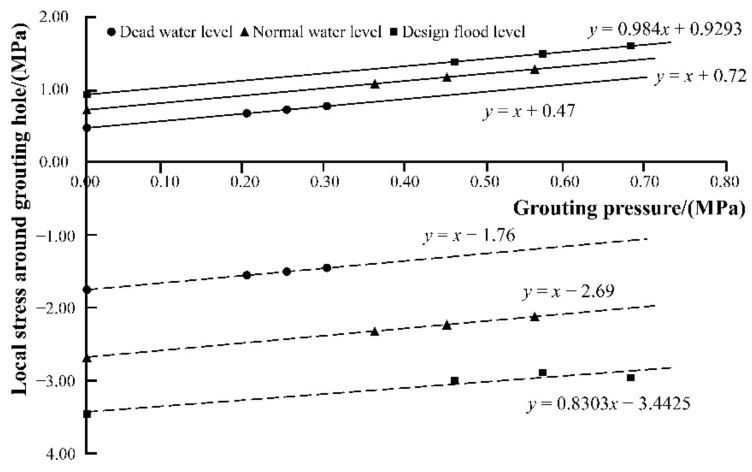
The relationship between extreme values of local stress and grouting pressure. *x* and *y*, respectively, refer to the grouting pressure and extreme value of local stress.

**Figure 16 materials-15-03520-f016:**
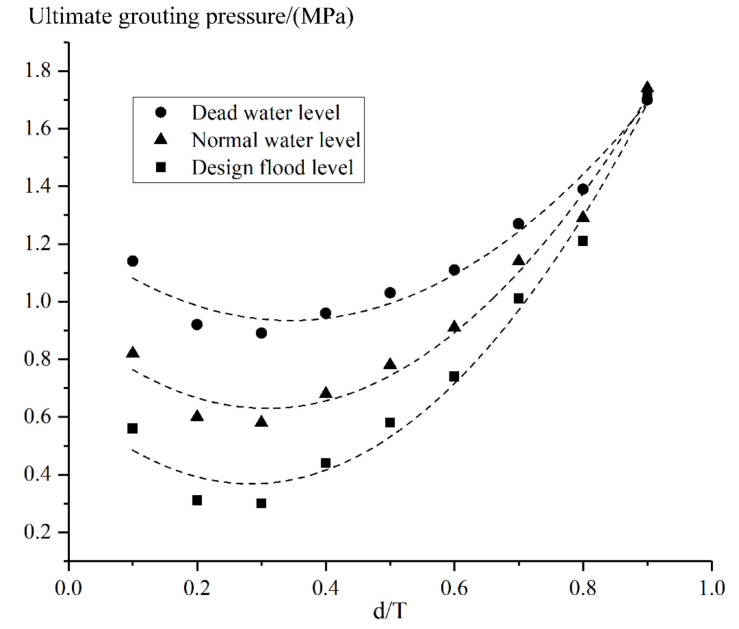
Ultimate grouting pressure at different water levels and grouting locations.

**Figure 17 materials-15-03520-f017:**
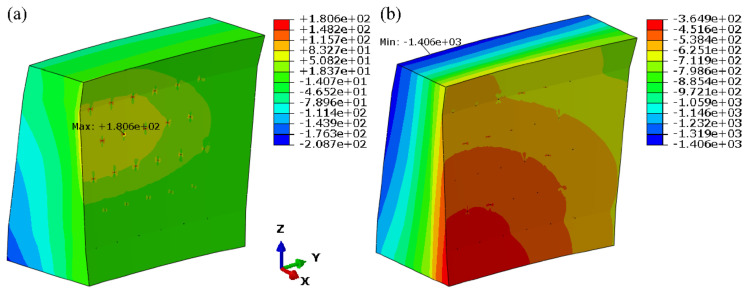
Stress contours of the area around the grouting test: (**a**) 1st principal stress; (**b**) 3rd principal stress. (unit: kPa).

**Figure 18 materials-15-03520-f018:**
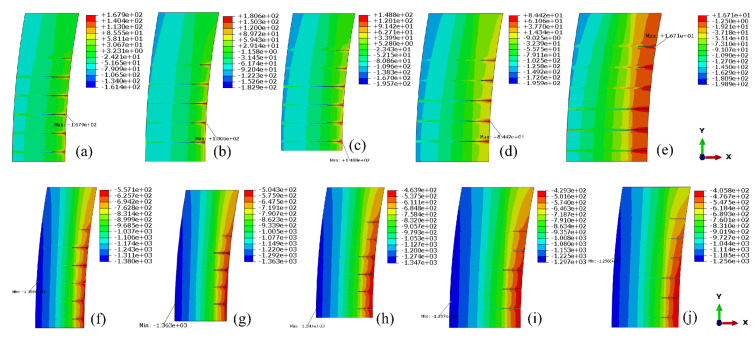
Stress contours of each section in grouting test area (Grouting pressure = 0.55 MPa): (**a**–**e**) 1st principal stress; (**f**–**j**) 3rd principal stress. (unit: kPa).

**Figure 19 materials-15-03520-f019:**
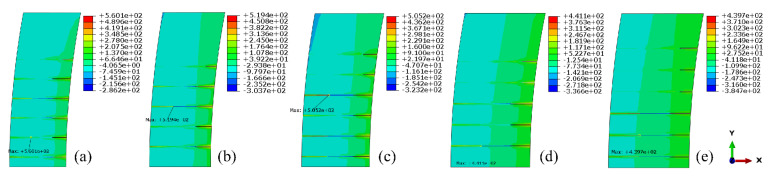
Tensile stress contours of each section in the grouting test area (Grouting pressure = 2.0 MPa): (**a**) 205.0m elevation; (**b**) 207.5 m elevation; (**c**) 210.0 m elevation; (**d**) 212.5 m elevation; (**e**) 215.0 m elevation. (unit: kPa).

**Figure 20 materials-15-03520-f020:**
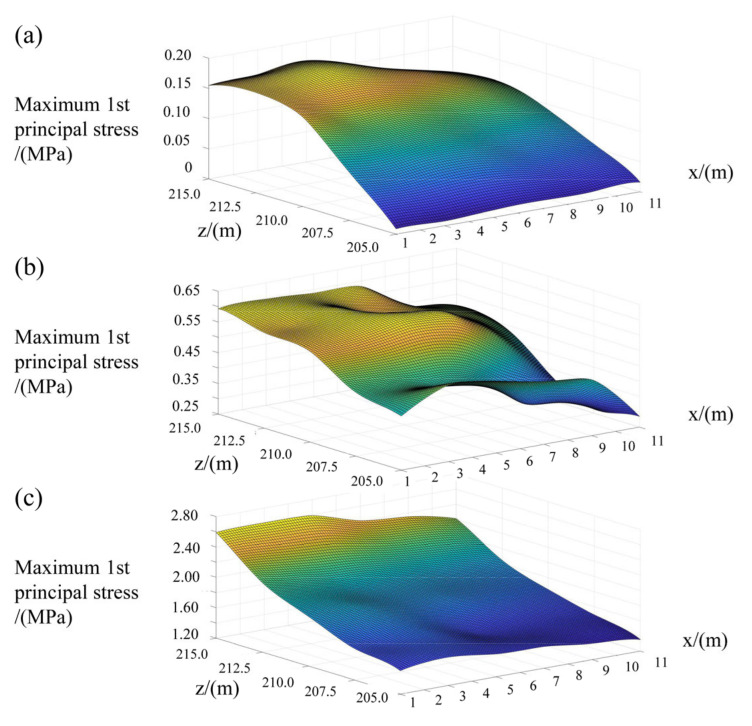
The 3D fitting surface of maximum tensile stress in the test area under different grouting pressure: (**a**) Grouting pressure equals to 0.55 MPa; (**b**) Grouting pressure equals to 2.0 MPa; (**c**) Grouting pressure equals to 5.0 MPa. *x* and *z*, respectively, refer to the distance between the grouting hole and crown cantilever, as well as the altitude of grouting holes.

**Figure 21 materials-15-03520-f021:**
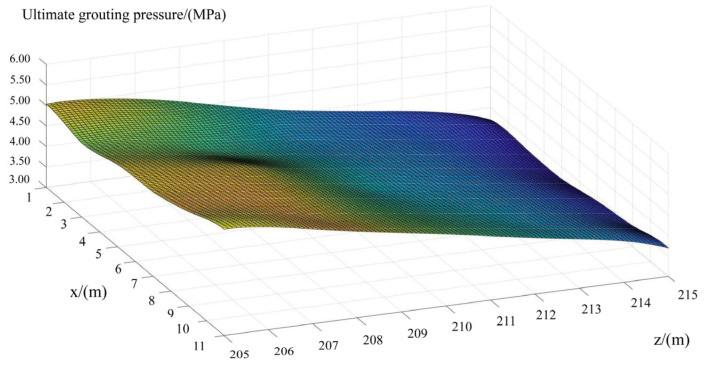
The 3D fitting surface of ultimate grouting pressure in test area.

**Figure 22 materials-15-03520-f022:**
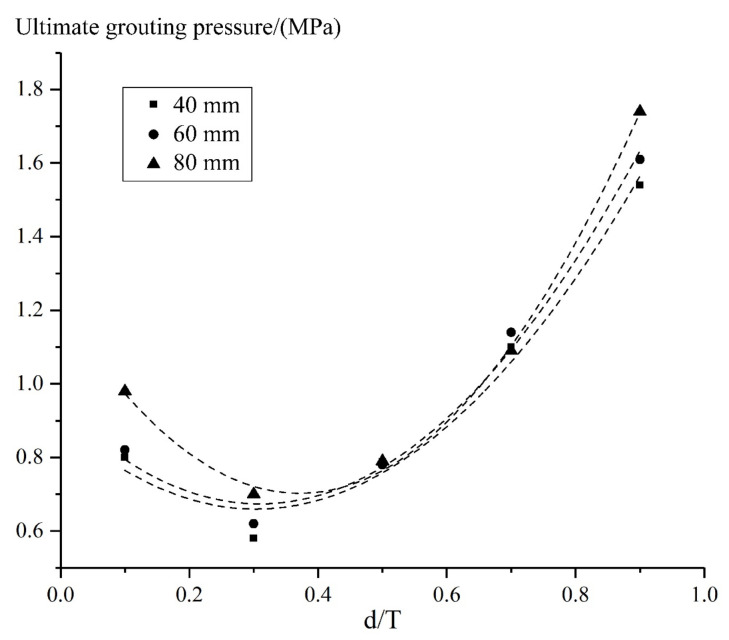
Effect of grouting hole diameter on ultimate grouting pressure.

**Figure 23 materials-15-03520-f023:**
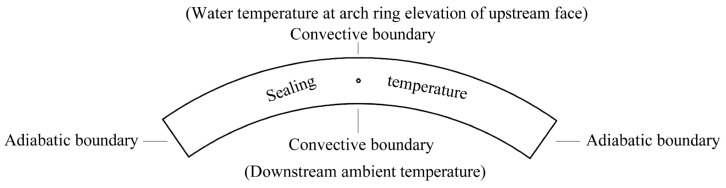
Temperature boundary conditions in the analysis of the arch ring temperature field.

**Figure 24 materials-15-03520-f024:**
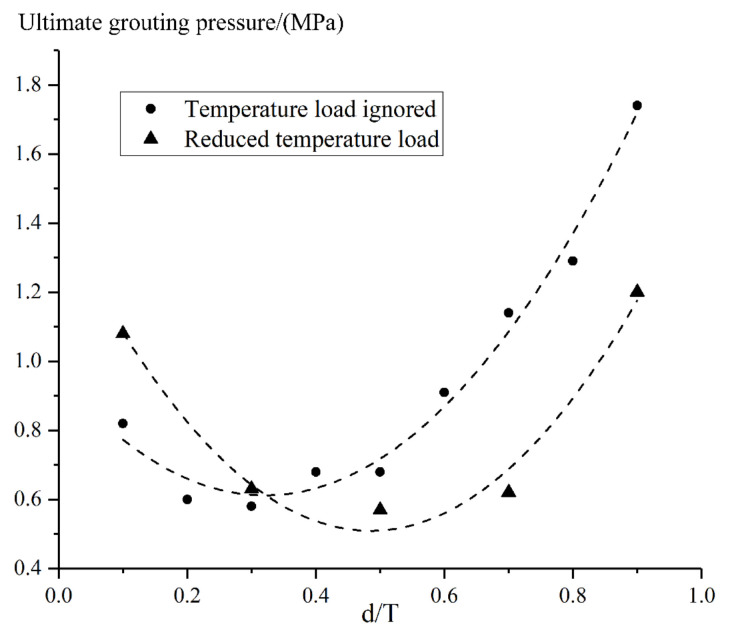
Effect of ambient temperature drop on the ultimate grouting pressure.

**Table 1 materials-15-03520-t001:** Main design parameters of the arch dam.

Parameter Type		Parameter	Unit
Characteristic water level	Dead water level	208.00	m
	Normal water level	226.00	
	Design flood level	231.22	
Temperature parameters	Arch dam closure temperature	15.4	°C
	Average monthly minimum temperature	3.0	
	Average monthly maximum temperature	27.3	

**Table 2 materials-15-03520-t002:** Materials parameters of dam body.

Materials	Dam Body	Impervious Panel	Impervious Curtain
Density/kg·m^−3^	2400	2400	2400
Elastic modulus/GPa	9.0	20.0	20.0
Poisson’s ratio	0.25	0.167	0.167
Coefficient of linear expansion/°C	7.0 × 10^−6^	10 × 10^−6^	10 × 10^−6^
Thermal conductivity/W·m^−1^·K^−1^	2.4	2.40	3.48
Specific heat capacity/J·kg^−1^·K^−1^	749	978	749

**Table 3 materials-15-03520-t003:** Statistics of freeze-thaw cycle times in the dam site area.

Times of Freeze-Thaw Cycle (*m*/*n*/*t*) ^1^						Freeze-Thaw Cycle Times in Different Ranges		
October	November	December	January	February	March	−5–+5 °C	−10–+5 °C	−15–+5 °C
-/-/- ^2^	2/-/-	12/8/-	11/16/-	9/12/-	6/-/-	25	36	0

^1^ *m*, *n*, and *t*, respectively, indicate that the freeze-thaw time conforming to −5–+5 °C, −10–+5 °C, and −15–+5 °C is *m*, *n* and *t*. ^2^ indicates that there is no freeze-thaw cycle within the corresponding temperature variation range.

**Table 4 materials-15-03520-t004:** Mortar deterioration degree corresponding to different operation years.

Service Time of the Dam	Equivalent Freeze-Thaw Cycle Times	Degree of Deterioration
0	0	0%
20	80	21%
40	160	39%
60	240	52%
80	320	63%

**Table 5 materials-15-03520-t005:** Allowable stress control index of the arch dam (unit: MPa).

Load Combination	Allowable Tensile Stress	Allowable Compressive Stress
Basic combination	1.2	6.0
Special combination	1.5	

**Table 6 materials-15-03520-t006:** Calculation schemes for sensitivity analysis of mortar deterioration.

Scheme	Degree of Deterioration	Elastic Modulus of Mortar	Comprehensive Elastic Modulus	Description
Ⅰ	0	2.88	9.0	Ignore mortar deterioration
Ⅱ	29.1%	2.04	8.16	Only 29.1% deterioration of mortar on downstream surface
Ⅲ	29.1%	2.04	8.16	29.1% deterioration of dam body
Ⅳ	39.1%	1.75	7.87	39.1% deterioration of dam body

**Table 7 materials-15-03520-t007:** Comparison of maximum displacement and stress with a different mortar deterioration degree.

Calculation Schemes	Displacement/(mm)			Principal Stress/(MPa)	
	Along the River	Vertical to the River	Settlement	1st Principal Stress	3rd Principal Stress
Scheme Ⅰ	16.1	3.76	3.97	1.41	−2.6
Scheme Ⅱ	17.4	4.08	4.12	1.43	−2.55
Scheme Ⅲ	17.7	4.12	4.15	1.46	−2.52
Scheme Ⅳ	18.5	4.27	4.29	1.51	−2.49

**Table 8 materials-15-03520-t008:** Maximum values of the first principal stress on dam upstream surface under different grouting positions and pressure (unit: MPa).

	Grouting Pressure/(MPa)	0.36	0.45	0.56
d^−1^/(m)				
10.0		0.179	0.178	0.177
7.5		0.180	0.178	0.177
5.0		0.180	0.179	0.178
2.5		0.181	0.180	0.180

^1^ Note: d refers to the distance between the grouting hole and the dam upstream surface.

**Table 9 materials-15-03520-t009:** Extreme values of local stress around the grouting hole under different grouting pressures.

d/T ^1^	Water Level	Dead Water Level (208 m)		
	Grouting Pressure/(MPa)	0.20	0.25	0.30
0.5	1st principal stress	0.67	0.72	0.77
	3rd principal stress	−1.56	−1.51	−1.46

^1^ Note: d refers to the distance between the grouting hole and dam upstream surface, and T refers to the thickness of the arch ring.

**Table 10 materials-15-03520-t010:** Sensitivity analysis results for grouting hole diameter.

	d/T	0.1	0.3	0.5	0.7	0.9
r/(mm)						
40		0.80	0.58	0.79	1.10	1.54
60		0.82	0.62	0.78	1.14	1.61
80		0.98	0.70	0.79	1.09	1.74

**Table 11 materials-15-03520-t011:** Sensitivity analysis results for reduced ambient temperature.

	d/T	0.1	0.3	0.5	0.7	0.9
Temperature						
Ignored		0.82	0.58	0.68	1.14	1.74
Reduced		1.08	0.63	0.57	0.62	1.20

## Data Availability

Not applicable.

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
