# Peer review of "A Method for Determining Ultimate Grouting Pressure for Reinforcement of Masonry Arch Dam with Mortar Deterioration: A Case Study"

_materials, 2022, doi:10.3390/ma15103520_

Round 1
Reviewer 1 Report
General comments:
The topic and purpose of the paper is of great interest to the dam engineering community. The approach and analysis are clearly described and the paper is very well written.
The paper presents a methodology to assess suitable grouting pressures, using equivalent continuum FE numerical modelling and taking into account different conditions. The influence of different factors such as grouting hole diameter, a decrease in ambient temperature and elastic modulus of dam material are analysed. Conclusions drawn from the study are used to propose improvement of current guidelines.
The approach is valid, although I believe that the discontinuous nature of masonry structures is more realistic simulated using discrete element techniques. These models may also include fluid flow through the discontinuities. However, there are situations in which the simpler models provide an effective analysis tool, given the difficulties of representing explicitly the discontinuities. This issue is properly mentioned by the authors at the end of the paper.
Specific comments:
- In line 189 please verify if the reference Yin et al is correct. Is it Yao et al?
- Figure 7 is missing.
- In lines 288 and 325, I believe that “Displacement vertical to the river” should be replaced by “ displacement perpendicular to the river”.
- In Figure 12 please indicate the location of the grouting hole (or grouting line) or the nodes where the concentrated forces are applied.
- In section 6.3 the authors should mention and discuss the hypothesis of using a more dilute slurry.
Author Response
Thanks for the reviewer’s comments. These comments are all valuable and very helpful for revising and improving our paper. We have studied these comments carefully and have made revisions which we hope meet with approval. Our response is as follows:
1.In line 189 please verify if the reference Yin et al is correct. Is it Yao et al?
Response: Referring to the existing test results in the published papers [20,22], the relationship between the mortar deterioration degree and the number of freeze-thaw cycles is obtained through generalization and fitting, so its correctness can be guaranteed. On the other hand, as explained in Page 06 line 185-line 191, the freeze-thaw cycle is the main reason for the deterioration of the mortar of the arch dam. Therefore, the above research conclusions on mortar deterioration are also reasonable in the case of this paper, and we do not think it is necessary to give other explanations to verify its correctness. The regression analysis of Eq. (2) is proposed by Yao et al, as shown in Page 06 line 192-line 197, and the corresponding reference is No. [20]. And the term of equivalent number of freeze-thaw cycles was proposed by Yin et al, as shown in Page 06 line 198-line 204, and the corresponding reference is No. [22].
2.Figure 7 is missing.
Response: We are very sorry for the problem of missing a figure. We have added the missed figure, as shown in Page 10 line 296. At the same time, we checked all figures through the manuscript.
3.In lines 288 and 325, I believe that “Displacement vertical to the river” should be replaced by “displacement perpendicular to the river”.
Response: We have revised the inappropriate expression according to the reviewer’s comments. The “vertical to the river direction” has been revised as “perpendicular to the river direction”, as shown in Page 10 line 306; and the “displacement vertical to the river” has been revised as “displacement perpendicular to the river”, as shown in Page 12 line 342.
4.In Figure 12 please indicate the location of the grouting hole (or grouting line) or the nodes where the concentrated forces are applied.
Response: We have revised this figure and added some explanation according to the reviewer’s suggestion, as shown in Page 13 line 376 and Page 13 line 369-line 374. The perpendicular grouting line was arranged on the arch crown section, which has been indicated in Figure 12. And the concentrated force was applied to the element nodes of the dam downstream surface firstly. Besides, because the stress distribution of the dam body is symmetrical to the arch crown section, only half of the grouting area was displayed.
5.In section 6.3 the authors should mention and discuss the hypothesis of using a more dilute slurry.
Response: Much thanks for the reminder of the reviewer. We have added discussions to explain the hypothesis of using a more dilute slurry, as shown in Page 21 line 591-line 593. The thinner the slurry used in reinforcement grouting, the less solid matter formed after solidification and the greater the shrinkage. Therefore, the effect of filling and consolidation will be worse if the slurry is too dilute.

Reviewer 2 Report
The article is interesting and should have the attention of the editorial board. Due to the peculiarity of the case study, very rare and important structure, it is an added value on the importance of the subject treated.
I have some comments regarding the quality and the soundness of the article as follow.
- In section 2.1, it is mentioned the presence of structural issues, i.e. cracking of mortar. For the sake of correctness the authors should better explain the structural pathology, highlighting the causes. It is important to accompany the explanation with data or tests.
- From figure 2, the masonry seems to be rubble masonry. It is necessary to specify the construction technology of the masonry.
- In figure 4, you show the damn before and after grouting, highlighting that the cracks and cavities were filled. It is not clear from figure 4. You should provide better information on the presence of cracks and cavities.
- Explain better this term: “corrosion effect on mortar”. Corrosion is generally used for metals!
- In equations 2, 3 and table 3, are used the same letters for different notations. You should rewrite the equations by maintaining different letters for different notations.
- Figure 5 is hardly readable.
- From figure 6, it seems that the grouting is vertical. It does not coincide with figure 8. If yes, is it a recommended method considering the double curvature of the damn? Wouldn’t it be more recommend a step by step reinforcement through grouting from bottom to top?
- The grouting pressure is not fully correct. It should be justified and schematized, where different situations could be encountered:
- Point radial pressure
- Surface pressure if a cavity is present in between two rigid blocks.
- The length “l”, in equations 11 and 12 should be justified by the cavities and not mesh. However, its length should be limited and not extended for the whole length.
Maybe the best modeling approach would require a multiscale approach. At a mesoscale, you determine the grout pressure based on a pore pressure numerical approach, varying the percentage of cavities and then applying this pressure based on a thermal expansion approach at the macroscale.
- The 3D model bears a conceptual error related to the phase construction. The most accurate approach is to consider the phase construction, therefore, the tensile stresses at the damn support that you have obtained would not appear. Those tensile stresses are nonrealistic for masonry damns and considering that double curvature damn is deliberately conceived to be without tension forces. A nonlinear static approach is at least required for the simulation ante grouting. A multi-leaf masonry wall simulation should be considered. Show the units of the stresses in the images.
- Additional explanation on graph 16 is required. What is notable is that the maximum pressure of grouting is when d/t approaches 1, or when the drilling hole riches the extreme of the upstream surface. The higher the grout pressure, the more efficient is to fill the cavities! However, based on figure 16, it means that we should drill close to the water surface so that the high pressure will not cause tensile cracks! On the other side, this zone, close to the water, is more prone to be vulnerable during this process, as the presence of water infiltration could cause more infiltration and put the process at risk. From figures 16 and 24, results show that under normal conditions, the lowest appliable pressure is at the middle of the damn thickness, generally speaking, where the cavities are expected to be higher as the least curated zone in a rubble masonry is the central thickness. A more comprehensive approach to the argument is required.
- From figure 18, it looks like the grouting holes are passing throughout the damn. Clarify it!
- Reconsider rewriting the discussion part! This part is very wordy and contains fewer scientific results and references.
- The references are not sufficient. I recommend some relevant literature in the following, particularly related to masonry modeling and FE modeling. I suggest further improving their literature with more relevant studies.
- 10.1016/j.jobe.2021.103929
- 10.3390/buildings11020071
- 10.3390/heritage4030135
Author Response
Thanks for the reviewer’s comments. These comments are all valuable and very helpful for revising and improving our paper. We have studied these comments carefully and have made revisions which we hope meet with approval. The reply is as follows:
1.In section 2.1, it is mentioned the presence of structural issues, i.e. cracking of mortar. For the sake of correctness the authors should better explain the structural pathology, highlighting the causes. It is important to accompany the explanation with data or tests.
Response: According to the reviewer’s suggest, we have highlighted the causes of mortar deterioration in more detail, as shown in Page 06 line 185-191. Because the arch dam is located in the temperate zone and the reservoir water quality is good, there is no long-term freezing and serious weathering. On the other hand, the temperature difference between day and night in winter at the dam site is large, so the freeze-thaw is deduced to be the main cause for mortar deterioration. The analysis of deterioration cause is mainly based on site inspection and reference to similar phenomena, such as getting wet and precipitating white solids on the downstream dam surface. Therefore, there is no data and tests for explanation, which we think is not necessary.
2.From figure 2, the masonry seems to be rubble masonry. It is necessary to specify the construction technology of the masonry.
Response: Much thanks for the reminder of the reviewer. We recognized that the hatch pattern and text interpretation in figure 2 may be ambiguous. Actually, the arch dam was stacked tidily by stone blocks and cemented by mortar. So we have adjusted the hatch pattern, as shown in figure 2. Besides, the “rubble masonry” is revised as “Mortar masonry”, as shown in Page 04 line 120.
3.In figure 4, you show the dam before and after grouting, highlighting that the cracks and cavities were filled. It is not clear from figure 4. You should provide better information on the presence of cracks and cavities.
Response: According to the suggestion of the reviewer, we have added more information about the cracks and cavities on dam body that have been grouted, as shown in Page 05 line 152-line 159. The phenomena of moisture and precipitation of white solids around the mortar layer between block stones are pretty obvious, which indicate that the mortar has deteriorated in a serious degree. Because the mortar layers are connected in the horizontal direction and not in the vertical direction, the cracks and cavities are mainly horizontal. Therefore, the grouting holes are arranged at the severely deteriorated part of the left downstream surface of the dam, and each row is arranged horizontally in the horizontal mortar layer.
4.Explain better this term: “corrosion effect on mortar”. Corrosion is generally used for metals!
Response: Much thanks for the reminder of the reviewer, we have revised the incorrect term. The original term “corrosion effect on mortar” has been revised to “effect on mortar deterioration”, as shown in Page 06 line 188-line 189.
5.In equations 2, 3 and table 3, are used the same letters for different notations. You should rewrite the equations by maintaining different letters for different notations.
Response: Thanks for the reviewer’s reminder. We have changed the notations in table 3 to other letters to avoid repetition with the equations. In detail, the original letter a, b, and c in table 3 are respectively changed to m, n, and t, as shown in Table 3 and Page 07 line 209-line 210. Besides, we also rechecked other notations through the manuscript.
6.Figure 5 is hardly readable.
Response: According to the reviewer’s comment, we added detailed explanations for Figure 5, as shown in Page 08 line 248-line 250. The two curves represent the change of reservoir water temperature with the increase of water depth under the conditions of temperature increase (July) and temperature decrease (January) respectively, in which the water depth of 0 is the temperature of water surface. Besides, we noticed that the figure is a little blurred, so we have changed the figure to improve its clarity, as shown in Page 08 line 253.
7.From figure 6, it seems that the grouting is vertical. It does not coincide with figure 8. If yes, is it a recommended method considering the double curvature of the dam? Wouldn’t it be more recommend a step by step reinforcement through grouting from bottom to top?
Response: Thanks for the reviewer’s comments. Actually, there are two different types of grouting is analyzed in this paper. Figure. 6 shows the simulation method for the influence of vertical grouting on the upstream dam surface stress, while Figure. 8 shows the three-dimensional finite element model established for analyzing horizontal grouting (adopted in actual construction). The two methods have little difference in the effect of grouting reinforcement, which should be selected according to the shape of arch dam and the site situation of the project. The explanation on the selection of grouting form is shown in Page 05 line 165-line 169. The grouting holes for arch dam reinforcement can only be drilled straightly and cannot be drilled along the curved shape of the arch dam. Due to the large curvature and narrow crest of the arch dam in this paper, vertical grouting holes cannot be arranged. Although the construction of horizontal holes is difficult, it can cover the whole area requiring grouting. Therefore, horizontal grouting is selected in this project. But at the same time, this paper also analyzes the problems of vertical grouting.
8.The grouting pressure is not fully correct. It should be justified and schematized, where different situations could be encountered: Point radial pressure. Surface pressure if a cavity is present in between two rigid blocks.
Response: Thanks for the reviewer’s comments. Actually, the term of “grouting pressure” is proposed by the technical specification for cement grouting of hydraulic structures [16]. In grouting, cement slurry is injected into the hole, which is a liquid. Therefore, the pressure generated by grouting is fluid pressure, which is equal in all directions. We think the grouting pressure described in this paper is reasonable.
9.The length “l”, in equations 11 and 12 should be justified by the cavities and not mesh. However, its length should be limited and not extended for the whole length.
Response: It is a discrete method to change the grouting pressure into concentrated forces through integration and then distribute them to the element nodes. Generally, vertical grouting is carried out step by step, so the total length of grouting hole is limited to the length of each grouting section. And the equivalent concentrated forces do act on the nodes of the mesh in the grouting section. We think this method is reasonable.
10.Maybe the best modeling approach would require a multiscale approach. At a mesoscale, you determine the grout pressure based on a pore pressure numerical approach, varying the percentage of cavities and then applying this pressure based on a thermal expansion approach at the macroscale.
Response: According to the calculation requirements proposed in the specification [15], the current FEM model and grid of the arch dam can meet the calculation accuracy. For this project, based on the results of the calculation model in this paper, the influence of mortar deterioration on dam stress can be explained, and the guiding significance for the construction of reinforcement grouting can be put forward. Therefore, multiscale model is not necessary.
11.The 3D model bears a conceptual error related to the phase construction. The most accurate approach is to consider the phase construction, therefore, the tensile stresses at the dam support that you have obtained would not appear. Those tensile stresses are nonrealistic for masonry damns and considering that double curvature dam is deliberately conceived to be without tension forces. A nonlinear static approach is at least required for the simulation ante grouting. A multi-leaf masonry wall simulation should be considered. Show the units of the stresses in the images.
Response: Thanks for the reviewer’s comments. Since the masonry arch dam is nearly 30 years away from the completion of construction, and there were no safety problems in the previous operation, the construction phase is not crucial to be considered, and the mortar deterioration after many years of operation is mainly analyzed, which is the focal point of this paper. On the other hand, the masonry material is regarded as homogeneous material, while the specific structure of block stones and mortar is ignored. Therefore, the occurrence of tensile stress at dam abutment and dam foundation is a general law of this model, which can be explained by stress concentration. In addition, for masonry arch dams, the specification proposes the allowable tensile stress, that is, a certain range of tensile stress area is allowed, as long as the allowable tensile stress is not exceeded [15]. It can be seen from the calculation results that the stress values of most areas of the dam body do not reach the allowable stress, and the dam body is mainly in the linear elastic stage, so the nonlinear static analysis method is not necessary to be adopted. At the same time, because the finite element model for homogeneous material in this paper meets the specification requirements and calculation accuracy, the multi-leaf masonry wall simulation is not required. And the units of the stresses and displacements have been added, as shown in Page 12 line 343, Page 12 line 350, and Page 13 line 378, Page 14 line 405, Page 15 line 408, Page 17 line 461, Page 17 line 467 and Page 13 line 475.
12.Additional explanation on graph 16 is required. What is notable is that the maximum pressure of grouting is when d/t approaches 1, or when the drilling hole riches the extreme of the upstream surface. The higher the grout pressure, the more efficient is to fill the cavities! However, based on figure 16, it means that we should drill close to the water surface so that the high pressure will not cause tensile cracks! On the other side, this zone, close to the water, is more prone to be vulnerable during this process, as the presence of water infiltration could cause more infiltration and put the process at risk. From figures 16 and 24, results show that under normal conditions, the lowest appliable pressure is at the middle of the dam thickness, generally speaking, where the cavities are expected to be higher as the least curated zone in a rubble masonry is the central thickness. A more comprehensive approach to the argument is required.
Response: Much thanks for the reminder of the reviewer, we have rechecked the calculation results and conclusions and we found out there was a mistake about the meaning of the notation d. In the original manuscript, the notation d in Figure. 6, Figure. 7 and Table 8 means the distance between grouting hole and dam downstream surface, and the notation d in Table 9, Figure. 16, Table 10, Figure. 22, Table 11 and Figure. 24 means the distance between grouting hole and dam upstream surface. We are very sorry for the mistake. In order to unify the meaning of the notation d, we change its meaning to the distance between grouting hole and dam upstream surface, as shown in Page 09 line 279, Page 10 line 296, Page 14 line 385, Page 15 line 425. So as shown in Figure. 16, the highest appliable grouting pressure is near the downstream surface of the dam, and the lowest appliable pressure is still at the middle of the dam thickness. The applicable grouting pressure near the upstream surface of the arch dam is slightly higher than that in the middle. This phenomenon can be explained by the distribution law of arch dam stress, as shown in Page 21 line 559-line 567. And we have recognized that the conclusions obtained according to Figure. 16 and Figure. 24 are not rigorous, so we have made further detailed explanation in the discussion and conclusion sections.
13.From figure 18, it looks like the grouting holes are passing throughout the dam. Clarify it!
Response: In the section stress contours in Figure. 18, although the value of stress along the direction of grouting hole is quite different from that nearby, it can only show that the application of grouting pressure will affect the distribution of local dam stress around the holes. However, since the stress value does not exceed the allowable compressive stress and allowable tensile stress proposed in the specification [15], the grouting will not lead to cracks penetrating to the upstream surface of the arch dam.
14.Reconsider rewriting the discussion part! This part is very wordy and contains fewer scientific results and references.
Response: Thanks for the reviewer’s reminder. The preconditions of part of the discussion have been supplemented to improve its scientificity and preciseness. Besides, we noticed that the original discussion part was very wordy, and we have removed some unnecessary words.
15.The references are not sufficient. I recommend some relevant literature in the following, particularly related to masonry modeling and FE modeling. I suggest further improving their literature with more relevant studies.
10.1016/j.jobe.2021.103929
10.3390/buildings11020071
10.3390/heritage4030135
Response: Much thanks for the reminder of the reviewer, we have studied three literatures recommended by reviewers and selected two of them to be added to the references according to their relevance, as shown in Page 24 line 721-line 724. And the explanation of masonry modeling is added, as shown in Page 10 line 300-line 303. In a mortar masonry arch dam, the block size is random, and there is no method to accurately model at present. The method of finite element modeling and simulation regarding block stones and mortar as homogeneous materials is reasonable and recommended by the specification [15]. At the same time, in future research we will pay more attention to the masonry modeling methods. In the 6.4 section we propose to treat block stones as quasi brittle materials and reflect the cementation of mortar through interface element instead of the simplified homogeneous material model used in this paper, as shown in Page 22 line 638-line 641.

Round 2
Reviewer 2 Report
- I agree with “In grouting, cement slurry is injected into the hole, which is a liquid. Therefore, the pressure generated by grouting is fluid pressure, which is equal in all directions”, but in line 280 you say “Thus, the equivalent concentrated forces acting on the element nodes at termination.” So it is not fully correct to model the fluid pressure as a concentrated load because it is impossible for a concentrated load to mimic the effect of the pressure in all directions. From a modeling point of view, the pressure is equivalent to thermal forces if in your model, you cannot apply ad hoc pore pressure forces.
- In equations 2 and 3 you use “ n is the number of freeze-thaw circles” and “[N] is the equivalent number of freeze-thaw circles”. I imagine that at a certain point, you say that: n=[N]???
- I think you have a misunderstanding on the phase construction. When dealing with very large objects, its construction requires a lot of time and during construction, the stresses and strains of a certain phase influence the stresses and strain of a successive construction phase. To implement a phase construction requires nonlinear analyses. In addition, the double arch damns are conceived to work under compression only. This means that theoretically, any tensile stresses are a consequence of the non-perfect geometry or due to boundary constraints of the numerical model. These aspects were required to be addressed by you. Your answer were very semplistics. You should elaborate further your work.
Author Response
Thanks for the reviewer’s comments. These comments are all valuable and very helpful for revising and improving our paper. We have studied these comments carefully and have made revisions which we hope meet with approval. The reply is as follows:
1.I agree with “In grouting, cement slurry is injected into the hole, which is a liquid. Therefore, the pressure generated by grouting is fluid pressure, which is equal in all directions”, but in line 280 you say “Thus, the equivalent concentrated forces acting on the element nodes at termination.” So it is not fully correct to model the fluid pressure as a concentrated load because it is impossible for a concentrated load to mimic the effect of the pressure in all directions. From a modeling point of view, the pressure is equivalent to thermal forces if in your model, you cannot apply ad hoc pore pressure forces.
Response: Much thanks for the reminder of the reviewer. Actually, the simplified method of "replacing hole with line" that transform grouting pressure in all directions to concentrated forces is only adopted in section 4.1 to discuss the effect of grouting on dam upstream surface. Except in this case, the grouting pressure is simulated as the real situation. Compared with the whole arch dam, the size of grouting hole is very small, and it is the most unfavorable situation for the stress of the dam upstream surface to convert the grouting pressure in all directions into concentrated forces pointing upstream through integration. Therefore, although this method cannot completely and truly reflect the grouting pressure in all directions, it is reasonable to analyze the adverse effect of grouting on the upstream dam surface stress. Because the upstream surface of the arch dam is close to the water, the effect of grouting on its stress needs to be focused. Thus, we think the method is rational to be applied in the study case. On the other hand, we also recognized that the lack of explanation of this method is easy to confuse the readers, so we have added more explanatory words, as shown in Page 09 line 277-line 280.
2.In equations 2 and 3 you use “n is the number of freeze-thaw circles” and “[N] is the equivalent number of freeze-thaw circles”. I imagine that at a certain point, you say that: n=[N]???
Response: n is the number of freeze-thaw circles during tests summarized by Yao et al [20], which is controlled by corresponding apparatuses. However, in the actual engineering operation environment, it is difficult to determine the number of times that buildings are subjected to freezing and thawing. Thus, the term of equivalent number of freeze-thaw circles is utilized in this paper to describe the relationship of freeze-thaw cycles and ambient temperature change [22]. In the calculation of this paper, the equivalent number of freeze-thaw cycles [N] is used to represent the number of freeze-thaw cycles n for the arch dam. We noticed that the difference of the two notations may be confusing to the readers, so we have added explanatory words, as shown in Page 06 line 195-line 202.
3.I think you have a misunderstanding on the phase construction. When dealing with very large objects, its construction requires a lot of time and during construction, the stresses and strains of a certain phase influence the stresses and strain of a successive construction phase. To implement a phase construction requires nonlinear analyses. In addition, the double arch dams are conceived to work under compression only. This means that theoretically, any tensile stresses are a consequence of the non-perfect geometry or due to boundary constraints of the numerical model. These aspects were required to be addressed by you. Your answer was very simplistic. You should elaborate further your work.
Response: Generally, the arch dam is constructed by sectional placing, which will become a whole only after the arch is sealed, and the materials of the dam body are basically in elastic state. For the structure composed of elastic materials, the results of stress calculation are the same under phase construction and one-time loading. And this paper mainly discusses the influence of different deterioration and grouting conditions on the dam stress, so it is not necessary to focus on the analysis of the stage of the construction period. On the other hand, in order to avoid the effect of dam self-weight on the displacement in the construction stage, the obtained model displacement with only gravity applied was cleared before exerting other loads. Therefore, Figure. 10 shows the dam displacement contours only under the action of water pressure, uplift pressure and temperature load, which is as same as the situation of considering phase construction. And the corresponding explanations are added and shown in Page 12 line 345-line 347. Besides, the stress distribution of arch dam can be explained by the trial load method. Theoretically, a double curvature arch dam can be regarded as a series of combination of horizontal arch rings and vertical beams, and all loads are borne by these arch rings and beams. Normally the arch rings only bear compressive stress. But the bottom of the beams will inevitably work under tensile stress, because the bottom of these beams is fixed with the foundation and the beams have the trend of displacement and rotation to the downstream under loads. Meanwhile, based on the article 6.2.6 of the design specification for stone masonry dam [15], when the finite element method is used to analyze the stress of masonry arch dam, a certain calculated tensile stress is allowed.
